# On the impact of dropsondes on the ECMWF IFS model (Cy47r1) analysis of convection during OTREC

Stipo Sentić[1], Peter Bechtold[2], Željka Fuchs-Stone[1,3], Mark Rodwell[2], and David J. Raymond[1,3]

[1]Climate and Water Consortium, New Mexico Tech, Socorro, NM, USA
[2]European Center for Medium Range Weather Forecast, Reading, UK
[3]Physics Department, New Mexico Tech, Socorro, NM, USA

**Correspondence:** Stipo Sentić (stipo.sentic@nmt.edu)

**Abstract.** The Organization of Tropical East Pacific Convection (OTREC) field campaign, conducted August through October 2019, focuses on studying convection in the East Pacific and the Caribbean. An unprecedented number of dropsondes were deployed (648) during 22 missions to study the region of strong sea surface temperature (SST) gradients in the East Pacific region, the region just off the coast of Columbia, and in the uniform SST region in the southwest Caribbean. The dropsondes were assimilated in the European Center for Medium Range Weather Forecast (ECMWF) model. This study quantifies departures, observed minus the model value of a variable, in dropsonde denial experiments, and studies time series of convective variables: saturation fraction which measures moisture, and instability index and deep convective inhibition which quantify atmospheric stability and boundary layer stability to convection, respectively. Departures are small whether dropsondes are assimilated or not, except in a special case of developing convection and organization prior to tropical storm Ivo where wind departures are significantly larger when dropsondes are not assimilated. Departures are larger in cloudy regions compared to cloud free regions when comparing a vertically integrated departure with a cloudiness estimation. Above mentioned variables are all well represented by the model when compared to observations, with some systematic deviations in and above the boundary layer. Time series of these variables show artificial convective activity in the model, in the east Pacific region off the coast of Costa Rica, which we hypothesize occurs due to overestimation of moisture content in that region.

## 1 Introduction

The Organization of Tropical East Pacific Convection (OTREC) field campaign was performed from Aug 5 to Oct 3, 2019, in the regions of far east Pacific and the Caribbean. The main goal of the field campaign (Fuchs-Stone et al., 2020; Raymond and Fuchs-Stone, 2021) is to study the convection in the east Pacific and the Caribbean. Specifically, since this area of the planet is sparse in observational data, the goal was to collect a data set to study the performance of weather models in this region, the interaction of local convection with passing easterly waves, and to test if some easterly waves originated in the region of the east Pacific. The flights performed in 22 missions, occurring over 22 separate days, were temporally randomly chosen so that easterly wave passage or formation could be randomly sampled, and the three regions were chosen so that passing easterly waves could be sampled on consecutive days. The areas of dropsonde deployment during OTREC are shown in Figure 1: the east Pacific boxes; B2 (12 flights), B3 (1 flight), and B1a (B1a off the coast of Colombia, 9 flights), and a box in the Caribbean,

B1b (7 flights, overlapping with B1a box flights). Figure 1 also shows the density, on a logarithmic scale, of observations used for assimilation in each 1 by 1 degree longitude-latitude box, summed over the whole OTREC observation period; the B3 box is a region with one flight and therefore the least data (see discussion later).

The East Pacific area of the tropics is well known for tropical cyclone genesis (e.g. Zehnder, 1991; Zehnder et al., 1999; Molinari and Vollaro, 2000), whether from initiation from local conditions, or from easterly waves which travel through the

region. Often utilised in study of these phenomena are the European Center for Medium-Range Weather Forecasts (ECMWF) analyses (e.g. Hersbach et al., 2020). As studies of easterly waves rely heavily on reanalysis data (Molinari and Vollaro, 2000; Hodges et al., 2003; Kiladis et al., 2006; Chen, 2006; Mekonnen et al., 2006; Ruti and Dell'Aquila, 2010; Serra et al., 2010; Janiga and Thorncroft, 2013; Rydbeck and Maloney, 2015), it is imperative to understand the influence of lack of free tropospheric data on these analyses in an observational data sparse region like the east Pacific.

A number of previous studies looked at the influence of dropsondes and radiosondes on the ECMWF analysis and reanalysis in other regions of the planet. For example, in the African Monsoon Multidisciplinary Analysis (AMMA) project (Agustí-Panareda et al., 2010) found that an extended radiosonde network decreased the large low level temperature and moisture bias in the analysis, thus affecting the model cloudiness, precipitation, and convection. They also found that easterly waves are weaker in the ECMWF analysis when additional dropsondes were not assimilated. Harnisch and Weissmann (2010) found that

assimilating dropsondes in regions near the cores of tropical cyclones had a positive impact on the analysis and track forecasts, during The Observing System Research and Predictability Experiment (THORPEX) Pacific Asian Regional Campaign (T-PARC) 2008. Schindler et al. (2020) found, during the North Atlantic Waveguide and Downstream Impact Experiment (NAWDEX), that additional dropsondes conservatively yet positively influence the forecast error of the ECMWF data. Chan et al. (2018) found that assimilating data for a tropical cyclone decreases the forecast errors by 13%, and produces a better

intensity and track forecasts. They found that assimilated humidity largely contributed to these improvements. In other models, for example, Feng and Wang (2019) studied the influence of dropsonde assimilation on modeling rapid intensification of hurricane Patricia (2015 hurricane season). Including dropsondes improves accuracy of outflow related parameters and thermodynamic analysis, and improves rapid intensification forecast. These and other ECMWF studies (e.g. Keil and Cardinali, 2004; Tompkins et al., 2005; Harnisch et al., 2011) have been performed outside the east Pacific region. Therefore, the OTREC

field campaign offers opportunity to explore similar questions in the east Pacific and Caribbean regions.

The OTREC field campaign is the first field campaign since EPIC2001 (e.g. Petersen et al., 2003; Raymond et al., 2004) which focuses on this region with an unprecedented amount of tropospheric observational data collected. One of the main instruments used during the OTREC field campaign were NRD41 dropsondes (Vömel et al., 2021) launched from the UCAR/NCAR Gulfstream-V aircraft, in addition to radiosonde launches from three land based sites: Limon and Santa Cruz on the east and

west coasts of Costa Rica, respectively, and Nuqui on the Pacific coast of Colombia. 648 dropsondes were successfully deployed during OTREC. The configuration of the drops during OTREC, i.e. high resolution dropsondes deployed from 13 km (EPIC2001 had sonde data up to 6.3 km), with a sampling frequency of 1 second, spaced horizontally about 1 degree from each other, allow for a more accurate assessment of fields sensitive to the horizontal data distribution, like vorticity and divergence, and other derived fields.

The European Center assimilated the dropsonde data in their operational analysis and reanalysis in the ECMWF model. This paper quantitatively assesses the impact of high resolution dropsonde data on the ECMWF analysis and modeling of tropical convection during OTREC. We perform an experiment with the ECMWF operational model assimilating dropsondes (labeled YDPS throughout the paper), and experiment with no dropsondes assimilated (labeled NDPS throughout the paper), and we compare these two experiments to observations (dropsondes). Note that radiosondes are always assimilated. Raymond and Fuchs-Stone (2021), using OTREC data, identified three parameters important for convection in the region: saturation fraction, instability index, and deep convective inhibition (DCIN). These parameters were found in previous research to be important in characterizing convection (Raymond and Sessions, 2007; Sessions et al., 2015; Sentić and Sessions, 2017; Gjorgjievska and Raymond, 2014; Raymond et al., 2014; Sentić et al., 2015; Fuchs-Stone et al., 2020; Raymond and Fuchs-Stone, 2021), and are explored in this paper; we also examine the time series of these parameters for convective and non-convective conditions.

Section 2 describes the data and variables used in this study. We look at basic fields in the model (zonal and meridional wind, moisture and temperature) on specified levels for both experiments, and compare them to observed values from the dropsondes, in section 3. We also estimate vorticity since the mid level values of vorticity are often used to diagnose the passage of easterly waves. In section 4 we look at the influence of cloudiness on the dropsonde assimilation, and compare the modeled and observed saturation fraction, instability index, and DCIN. Finally, section 5 assesses the ECMWF operational model (since it assimilated the dropsondes) and time series of thermodynamic variables obtained from it to measure the performance of the model for convective and non-convective regions. We summarize the discussion in section 6.

## 2   Data and methods

This study exclusively uses OTREC dropsonde in situ observations (Earth Observing Laboratory and Voemel, 2019; Fuchs-Stone et al., 2020; Raymond and Fuchs-Stone, 2021; Vömel et al., 2021) of zonal wind, meridional wind, potential temperature, mixing ratio, and pressure. To grid the data from an irregular to a regular grid, which we use as a 3D proxy of the dropsonde observations, we use a 3D-variational (3D-var) analysis calculated by a penalty function minimization (Raymond and López Carrillo, 2011; López Carrillo and Raymond, 2011). This produces a gridded data set where the interpolated values in between dropsondes satisfy the mass continuity equation. This 3D-var approach was used in many previous studies of convection (Gjorgjievska and Raymond, 2014; Fuchs-Stone et al., 2020; Raymond and Fuchs-Stone, 2021). We use the 3D-var gridded data for calculating derived fields, like the moisture convergence as a measure of convective activity in section 5.

Basic and derived model analysis fields used in this study come from the ECMWF Integrated Forecast System (IFS, version Cy47r1) operational model. Control variables for assimilation are vorticity, divergence, temperature and relative humidity. Studies like Xie and MacDonald (2012) show that the choice of vorticity and divergence as control variables produces smoother final analysis, compared to using zonal and meridional winds, or the stream function and the velocity potential. Model horizontal resolution is TCo1279, cubic octahedral grid with maximal total wavelength 1279, which is around 9 km; we retrieve the data on a 0.1 degree regular longitude latitude grid on 18 pressure levels. The model is run over one cycle from Aug 7 to Sep 30, 2019, in a dropsonde assimilated mode (labeled YDPS throughout this paper), and dropsondes *not* assimilated mode

(labeled NDPS throughout this paper). By comparing the model output in the YDPS and NDPS experiments with dropsonde observations, we can quantify how large the errors, or departures of the model from observation, are in the ECMWF analysis when such a rich in-situ observational data set is not available for assimilation. The departures of the analysis model state from the observations are defined by:

$$x = x_{observation} - x_{model}, \tag{1}$$

where $x$ is one of the basic fields: zonal wind ($u$), meridional wind ($v$), water vapor mixing ratio ($q$), and potential temperature ($\theta$). Note that the positive and negative values of departure denote overestimation and underestimation in the model, respectively. Reduction of the departures after assimilating the dropsonde data means that the model is able to make use of the data and that errors are reduced. The model fields are interpolated to the location of each observation via a 4D-VAR, so that departures from each observation can be computed. We analyse data only used in both experiments, and model data which have observations. Therefore, the results are for the reported significant levels (i.e. where the observation value changes significantly), but we show departures for mandatory levels up to the 200 hPa pressure level.

The ECMWF model uses a varied set of observational data for 4D-variational assimilation (Magnusson et al., 2019, 2021). In situ data used for assimilation is received from: drifter and moored buoys (surface temperature, pressure, wind, ocean temperature and salinity), SYNOP observations (surface pressure, wind, temperature, humidity, snow), SHIP/METAR (surface pressure, wind, and temperature), TEMP/TEMPSHIP/DROPSONDES (temperature, humidity, and wind profiles), aircraft (temperature, humidity, wind), profilers (wind profiles), ARGO/XBTS/CTDs (ocean temperature and salinity profiles), NexRAD (precipitation). Remote sensed satellite data (METOP B/C, AQUA, NPP, NOAA-20, METEOSAT, GOES, HIMAWARI, DMSP, COSMIC, SPIRE, Sentinel-6A, FY, FY-3, GRACE-C, HY-2B, JASON, SARAL, CRYOSAT, MTG) has been used for assimilation in both cloud-free and cloud covered regions. Temperature, humidity, and winds are derived via 4D tracking from passive infrared radiances, and passive microwave radiances for both clear sky, and all sky conditions. Wind is also derived from satellite atmospheric motion vectors, while temperature and humidity can be retrieved from GPS radio signal phase delays. Scatterometers measure sea surface winds and soil moisture, while altimeters measure sea surface height and significant wave height. In the NDPS experiment described above, only the DROPSONDE data source is turned off in the model assimilation algorithm.

Despite the fact that vorticity and divergence are control variables, we cannot compare them to the observed zonal and meridional winds directly. Therefore, we resort to estimating the divergence and vorticity, and their departures, for observations and the model using the point data of modeled and observed zonal and meridional winds on each pressure level. We linearly interpolate the wind data on a regular grid from which we calculate vorticity and divergence. Although the point data we get from the above procedure is produced in a 4D variational procedure with point data separated in time, we assume the drops were done simultaneously for the vorticity and divergence estimation. The OTREC lawnmower flight patterns were designed to minimize time skew in calculating vorticity, divergence, and other derived fields.

Furthermore, we use thermodynamic variables shown in previous research (Fuchs-Stone et al., 2020; Raymond and Fuchs-Stone, 2021) to be significant in diagnosing the development of convection: saturation fraction, instability, and DCIN. Satu-

ration fraction is defined as precipitable water divided by saturated precipitable water, which is a measure of column relative humidity. Instability index, defined as saturated moist entropy between 1 and 3 km minus the saturated moist entropy from 5 to 7 km, is a measure of the mid to low tropospheric moist convective stability. Contrary to intuition, the lower the instability index, the more conducive the environment is to deep convection. This was found in many studies of convection, both isolated convection (Raymond and Sessions, 2007; Sessions et al., 2015; Sentić and Sessions, 2017) and organized convection like hurricanes and the Madden-Julian Oscillation (Gjorgjievska and Raymond, 2014; Raymond et al., 2014; Sentić et al., 2015). Deep Convective Inhibition (DCIN), defined as the mean saturated moist entropy from 1.5 to 2 km minus the mean moist entropy from 0 to 1 km in the boundary layer, was also shown to play a significant role in convective development (Sentić et al., 2015; Fuchs-Stone et al., 2020; Raymond and Fuchs-Stone, 2021). Convection develops when DCIN is small or negative.

ECMWF operational analysis forecast initialized at 12 UTC is used for calculating the diurnal cycle of the above variables for non-convective, and convective cells averaged into 1 by 1 degree boxes. We define non-convective and convective cells as having moisture convergence less than 0.5 kW m$^{-2}$ and greater than 2 kW m$^{-2}$, respectively, where moisture convergence is defined as:

$$\Delta_m = -\int_{p_s}^{p_t} \nabla \cdot (q\mathbf{v}) dp, \tag{2}$$

where $p_s$ and $p_t$ are surface and pressure at the top of the dropsonde path, respectively, $q$ is the mixing ratio, and $\mathbf{v}$ is the horizontal wind vector. For selected cells the diurnal cycle is calculated to produce a mean diurnal cycle for both non-convective and convective cells.

Also used are NOAA GOES-R 16 channel 14 infrared temperature (Schmit et al., 2017), averaged in a 0.5 by 0.5 degree box around each dropsonde, to quantify cloudiness (see section 4).

## 3 Departures of basic and derived fields

This section examines the departures of ECMWF zonal and meridional winds, water vapor mixing ratio, and potential temperature from actual dropsonde values, for the two experiments with (YDPS) and without (NDPS) dropsondes, from the ECMWF operational analysis. First, horizontal departures are examined as a function of longitude and latitude (section 3.1) for a select pressure level, i.e. 700 hPa. Next, vertical profiles of departures are shown as a function of pressure (section 3.2). Understanding the locations of the largest departures can inform us about systematic model errors and random errors in modeling convective systems. A special case in box B3 is also examined; a single OTREC research flight done in tandem with the NOAA P3 mission into the precursor convection of tropical storm Ivo (2019 hurricane season). Finally, we examine estimates of vorticity and divergence for convective regions in section 3.3.

### 3.1 Horizontal distribution of background departures

The lower troposphere above the boundary-layer is critical for the development of convection. We found the largest departures (defined in equation 1) at the 700 hPa level. Figure 2 displays the mean departures for zonal and meridional winds, while

the departures for specific humidity and temperature are displayed in Figure 3. The top row of panels shows mean dropsonde observations, middle row departures of the YDPS experiment, and bottom row departures of the NDPS experiment for zonal wind on the left and meridional on the right, for Figure 2. Both zonal and meridional wind have smaller departures in the YDPS experiment than in the NDPS experiment for all boxes. In the B3 box region, however, the departures are large in the NDPS experiment. This special case was flown on Aug 18, when the NCAR Gulfstream-V flew in conjunction with the NOAA P3 operational aircraft to investigate the precursor convection of tropical storm Ivo. Departures in box B3 are statistically significant which is obvious when mean departures are plotted for each individual flight (not shown)—the mean departures in box B3 are larger than the mean departures for all other individual flights. iThis suggests that the model might need more data in the pre-storm phase of a developing tropical cyclone to ingest and produce better initial conditions, at least in the East Pacific tropical region. The regions of boxes B2, B1a and B1b seem less sensitive to absence of dropsondes in the denial experiments since they do not show drastic changes in departures between the YDPS and NDPS experiments.

Figure 3 shows a similar picture. For both water vapor mixing ratio and temperature, the NDPS experiment shows larger values of departures compared to the YDPS experiment. However, in the B3 box we do not see larger than average departures in thermodynamic values; only the zonal and meridional winds suffered large departures in B3 box mentioned above. This suggests that in the absence of dropsonde observations, assimilation of wind fields from satellite data and other sources could be refined in organised convection conditions as exhibited in the case of precursor to tropical storm Ivo.

To better understand the conditions in which the special case in box B3 developed, we plot the mean zonal wind, meridional wind, mixing ratio, and potential temperature for each flight individually in figure 4. We subtract the mean water vapor mixing ratio and potential temperature taken over all OTREC dropsondes to get anomalies of both. Cases in the B1 (both B1a and B1b) box and the B2 box are shown in green and blue, respectively, while the special case in box B3 is shown in red. The special case shows very weak mean winds below 700 hPa compared to other cases, with opposite zonal winds in the layer 400 to 700 hPa. We also see an anomalously high moisture content in the troposphere and a drier boundary layer, with a warm anomalous layer below 500 hPa. A ECMWF report (Bechtold et al., 2012) suspects that the atmospheric motion vector algorithm for satellite wind retrievals (via 4D feature tracking) could have large errors in weak and divergent wind conditions, which seems to be happening in the special case. Furthermore, the anomalous higher water vapor mixing ratio could be contributing to large errors in satellite assimilating as it was shown in previous research (Geer et al., 2019).

## 3.2 Vertical structure of departures

Figure 5 shows the mean and standard deviation of departures from the observations calculated over all the dropsonde observations and all the regions from figure 1, for zonal wind (figure 5a), meridional wind (figure 5b), mixing ratio (figure 5c), and potential temperature (figure 5d), for both the YDPS (black lines) and the NDPS experiment (red lines). The zonal and meridional wind both show smaller departures and standard deviations in the YDPS run, with the NDPS experiment having a small negative bias (see equation 1) in the layer from about 800 to 500 hPa for the zonal wind, and the meridional wind having a positive bias in the layer from about 900 to 600 hPa. As absolute differences these departures are not large considering, for example, that the relative error is about 25% and 40% for the YDPS and NDPS experiment zonal wind mean of about 2 m s$^{-1}$

at about 700 hPa, respectively (not shown). The mixing ratio (figure 5c) shows a slight dry model bias throughout the tropo-
sphere with a relative error less than 4% to 6 % between the observations and the YDPS and NDPS experiments, respectively
(not shown). Similarly for potential temperature, there is a slight cold model bias in the model for both the YDPS and NDPS
experiment. In all cases, the inclusion of the soundings contributed to decreased spread of the departures as seen in the smaller
range of the standard deviation. Both in the zonal and horizontal wind, largest departures occur around 700 hPa in the YDPS
and NDPS experiments. General tropical wind errors are large at 700 hPa, reflecting errors in the large scale tropical circula-
tion. Recent improvements in the ECMWF operational model (version Cy47r3), as reported in an official ECMWF newsletter
(Forbes et al., 2021) show improvements in wind departure errors.

The departures above were computed by averaging over all the research flights. Here we focus on the special case in box
B3, addressed in section 3.1. The horizontal averages in Figures 2 and 3 show that we find largest departures in zonal and
meridional wind in box B3. Figure 6 shows the averages similar to Figure 5 but only for the research flight in box B3. The
zonal wind, meridional wind, and temperature, show large departures in the NDPS experiment, and including the dropsondes
drastically reduces the mean departures and the standard deviation of the departures in the YDPS experiment. As noted in the
previous section, this improvement in wind departures is probably due to better winds from dropsondes, as the satellite winds
derived with atmospheric motion vector algorithms are suspected to have large errors in weak wind and divergent conditions
(Bechtold et al., 2012). Local jets like the Tehuantepec jet (Chelton et al., 2000a, b) could have contributed to the strongly
divergent flow conditions in the B3 box.

In summary, considering the amount of satellite data used for assimilation in absence of dropsonde data, departures are
small on average as the observing system is well constrained and determined by infrared and microwave satellite observations,
and conventional observations, especially over sea as over land satellite channels with strong sensitivities close to the surface
cannot be assimilated (Geer et al., 2017, 2018, 2019). However, in weak wind and divergent flow conditions as in tropical
cyclone environments, as exhibited in the B3 box, departures can be large. The next section derives estimates of vorticity and
divergence from the dropsonde data to understand how the departures in the basic wind fields affects the departures in these
derived fields.

### 3.3 Vorticity and divergence estimates

Relative vorticity and divergence are important for study of convection and identifying the passage of easterly waves. Therefore,
it is necessary to estimate these variables and gauge how assimilating dropsonde data contributes to the integrity of the analysis.

As model departures for vorticity and divergence are not available from the ECMWF system, unlike the basic fields from the
previous sections (winds, mixing ratio, and potential temperature), we estimate these fields from the dropsonde observations
and the denial experiments by first linearly interpolating the model and observational point data onto a regular grid. The vor-
ticity and divergence are then calculated using finite differences, and the calculation is confirmed by comparing the circulation
around the perimeter of interest calculated using the vorticity and the circulation theorem:

$$\Gamma = \oint_{\delta S} \boldsymbol{v} \cdot d\boldsymbol{l} = \int \int_{S} \zeta dS \qquad (3)$$

where $\Gamma$ is the circulation, $\boldsymbol{v}$ is the wind speed, $S$ is the area over which circulation is being calculated over, and $\zeta$ is the vocticity. The circulations computed from both methods agree to within 10% in the lowest 500 hPa giving us confidence in the derived vorticity and divergence. However, larger errors appear above 500 hPa, hence those levels are excluded from the

following vorticity analysis. During an OTREC research flight the dropsondes were dropped over a span of 6 hours, but for the purpose of estimating the vorticity and divergence we assume that the dropsondes have been dropped at the same time. However, the OTREC flight patterns have been designed to minimize the time skew in calculating vorticity and divergence. For consistency with previous research (Fuchs-Stone et al., 2020), the divergence and vorticity are averaged in the regions of developing and decaying convection as tabulated in Table 1 from Fuchs-Stone et al. (2020).

Figure 7 shows the vorticity and divergence departures calculated for the YDPS and the NDPS experiments, Figure 7a and 7b for all the cases from Fuchs-Stone et al. (2020), and Figure 7c and 7d, for the special case in the B3 box. The B3 box case (figure 7c and 7d) does not show a standard deviation line because it is a single average over the west half of the B3 box. Both the YDPS and NDPS experiments in Figure 7a and 7b produce reasonably small mean vorticity departures, but assimilation of dropsondes improves the standard deviation of departures. Assimilation of dropsondes also benefits the

divergence calculation in the lower troposphere where there is also a reduction of the standard deviation of the departures and even the mean of departures. In this estimate, the divergence departures increase both in the mean and the standard deviation for levels higher than 700 hPa even for the YDPS experiment. Since the absolute values of vorticity and divergence are between 0 and 0.02 ks$^{-1}$ (not shown), the departures are relatively small (10-45% below 700 hPa). However, there seems to be a systematic underestimation of the divergence below 700 hPa for both the YDPS and the NDPS experiments. The B3

box case (figure 7c and 7d) shows large differences in the vorticity departures between the YDPS and NDPS experiments. Assimilating dropsondes (YDPS experiment) decreases the departure value for the layer between 500 and 900 hPa, however both experiments show larger departures below 900 hPa. The divergence shows similar departures for both experiments for the B3 box.

In summary, we find that there is a small difference between the estimated vorticity and divergence for the YDPS and the

NDPS experiments, with a systematically larger departure of divergence for both experiments. We also find that for the special case, B3, departures of vorticity are much smaller when dropsondes are assimilated.

## 4 Cloudiness effects and thermodynamic variables

In the last decade, operational weather models like the ECMWF model have been transitioning towards assimilating all sky microwave radiance data, as opposed to using only clear sky infrared radiance data used earlier (Geer et al., 2017, 2018, 2019).

We use infrared satellite imagery as a simple measure of cloudiness, since values of infrared brightness temperatures are lower for cloud covered regions. Previous studies did find influence of higher cloud cover (and moisture) to influence the analysis and forecast in the ECMWF model (Geer et al., 2019). This section examines the influence of cloudiness on the assimilation of dropsondes during OTREC, and we look at derived fields found useful in studying convection, defined in section 2; saturation fraction, instability index, and DCIN.

The OTREC field campaign sampled the tropospheric environment in both convective, cloudy, and non-convective, cloud-free, conditions. Therefore, it is possible to assess the influence of dropsonde assimilation in the ECMWF model for cloudy and cloud free regions. For each dropsonde spot we calculate a measure of the tropospheric departure at the location by vertically integrating each variable departure:

$$x^* = \frac{1}{p_s - p_t} \int\limits_{p_s}^{p_t} |x| dp \qquad (4)$$

where $x$ is the departure (defined in equation 1) for any of the basic variables of interest ($u$, $v$, $T$, and $q$), $p$ is pressure, and $p_t$ and $p_s$ are the top and surface pressure for each individual dropsonde, respectively. For each dropsonde location a corresponding infrared temperature is averaged as a measure of cloudiness, with cloud free regions defined by large values of infrared temperature, and cloudy regions defined with lower infrared temperatures. The vertical departures and the cloudiness proxy are then compared. As an example, Figure 8 shows $u^*$ for two soundings. A smaller vertically integrated departure is shown for a random dropsonde shown in Figure 8a, compared to more than double value shown for a previous sounding in Figure 8b. In these examples there is a difference in the vertical distribution of the departure of $u$ but that is not captured in the single number $u^*$. We compare the vertically integrated departure with infrared temperature over each dropsonde position. We use a 0.5 deg by 0.5 deg box to average the infrared temperature as a measure of cloudiness; we found that the results are not very sensitive to the size of this box. Furthermore, this cloudiness measure is very simple and does not address the vertical distribution of clouds, which is left to future, more detailed, studies.

Figure 9 shows the vertically integrated departures for u, v, q, and $\theta$ versus satellite infrared brightness temperature, for the YDPS experiment in black, and the NDPS experiment in red. A linear fit is applied to each scatter for each variable and the correlation coefficient and fit coefficients are shown in the corner of each figure panel. Also, squares are drawn for the 32 dropsondes from the B3 special case. First, for high values of infrared temperature, e.g. greater than 280 K (cloud free conditions), there is a clustering of points for each variable, with relatively small $x^*$, which means that in fair weather the ECMWF model has small departures for all variables. Second, for cloudier conditions, i.e. infrared temperature less than 280 K, there is significantly more scatter in all variables, especially in $u$ and $v$, as indicated by the slope of the fitted lines. The slope of the lines fitted in the wind scatter plots are larger than for the water vapor mixing ratio and the potential temperature. $q^*$ and $T^*$ have similar vertical departures for both cloudy and cloud free conditions, and have weaker scatter and smaller departures than the winds. We also see that the B3 box special case values (denoted with squares around the points), show the largest vertical departures in the NDPS experiment for all variables (most values appear above the red line in all panels). All the values for the special case show improvements in the vertical departures for the YDPS experiment. Apart from the line slope, we can infer the performance of assimilating the dropsondes in the model by observing the fit offset and correlation coefficients. The offset is larger for all the variables for the NDPS experiment, especially for $u^*$ and $v^*$. In general correlation coefficients are small. However, the wind correlation coefficients are larger compared to temperature and moisture, showing stronger wind departure dependence on cloudiness. Also, the correlation improves for all the YDPS experiments in all variables except the water vapor mixing ratio which seems to worsen. The robustness of the correlation between the integrated wind departures and

the infrared brightness temperature suggests an influence of cloud cover on satellite wind assimilation algorithms. Perhaps the OTREC data set can be used to improve these algorithms in the east Pacific region.

Figure 10 shows the scatter plots of saturation fraction (panels a and b), instability index (panels c and d), and DCIN (panels e and f), calculated from the dropsonde observations (OBS) and the YDPS experiment on the left, and between the YDPS and NDPS experiments on the right. The correlation coefficient between each variable pair is listed in each panel. As shown in the previous section, moisture and temperature fields are not very sensitive to the assimilated dropsondes as much as the wind field is. Therefore all the variables are well constrained in both YDPS and NDPS experiments. For example, an 8 J/K/kg and 4 J/K/kg change in the instability index and DCIN would correspond to about 1.2 K change in the temperature, respectively, with a 0.06 change in saturation fraction would correspond to about 1 g/kg change in water vapor mixing ratio. The difference in observed and modeled instability index has an average of -1.0±4.7 J/K/kg for the YDPS experiment, and -1.6±6.4 J/K/kg for the NDPS experiment. For DCIN those numbers are 0.6±9.6 J/K/kg and -0.4±11.4 for the YDPS and NDPS experiments, respectively, with such a large standard deviation due to the slant of the scatter. Saturation fraction has 0.00±0.06 and -0.01±0.07 for the YDPS and NDPS experiments, respectively. From these numbers we can see that the deviation between the model and observations falls below the values listed above. Furthermore, the YDPS experiment shows higher correlation coefficients with instability index showing the largest improvement when dropsondes are assimilated. DCIN seems to deviate from observations, with the YDPS and NDPS experiments both giving stronger DCIN compared to OBS. This suggests that boundary layer values of moisture deviate from the observed, probably due to model boundary layer processes, and the difficulty of assimilating boundary layer observations from satellite (Geer et al., 2017, 2018, 2019). In summary, the variables we find useful in modeling and understanding convection, i.e. saturation fraction, instability index, and somewhat DCIN, seem to be well represented by the ECMWF model in all conditions with instability index showing most improvement when dropsondes are assimilated in the model, this gives confidence in using these fields in reanalysis data.

## 5   Diurnal variability for convective and non-convective regions

A lot of information is derived from the above departure analysis. However, to evaluate the impact of the assimilated data on the performance of the ECMWF operational analysis in modeling physics and convection characteristics during OTREC, we examine time series of thermodynamic variables mentioned in the previous section. The performance of the model in the time domain can give us insight in whether the model overestimates or underestimates processes important for the development of convection in the model, and therefore has an influence on the forecast of convection and other phenomena like easterly waves.

We use the 3D-var analysis explained in section 2, and ECMWF operational analysis data on the days of the individual research flights. The 3D-var analysis of the observational dropsonde data is used to calculate average moisture convergence in 1 by 1 degree boxes which is used to define convective (moisture convergence greater than 2 kW m$^{-2}$) and non-convective (moisture convergence less than 0.5 kW m$^{-2}$) regions in the OTREC flight domains. For each of those cases the diurnal cycle of variables is taken at the location of the convective or non-convective one by one degree boxes, and averaged to obtain a mean diurnal cycle of variables for convective and non-convective regions. Figure 11 shows the mean diurnal cycle of ECMWF

model moisture convergence, infrared temperature (from observations), saturation fraction, instability index, and DCIN, for convective (red) and non-convective regions (black). Plotted are convective regions total mean diurnal cycle (red), which is decomposed into boxes B2 (blue) and B1a (green) for the reasons mentioned below. The diurnal cycles are calculated from 1047 individual cases for the non-convective case, 99 for the convective case, 33 in the B1a box, and 66 in the B2 box. The 6 B1b convective cases are excluded because the mean calculated from these is not statistically significant for such a low case

count. Vertical thin black lines bracket the period when the Gulfstream-V flew.

There is a stark difference between the mean diurnal cycle of convective regions (red line) and non-convective regions (black line). Convective regions show much lower values of instability index and DCIN, and at the same time show larger values of saturation fraction, compared to non-convective regions. In convective regions, the diurnal changes of saturation fraction, instability index, and DCIN, agree with previous research, namely that instability index and DCIN decrease between

330 0 and 12 UTC, and that saturation fraction increases. The increase of saturation fraction is in agreement with increase of moisture convergence, peaking around 12 UTC (moisture convergence is associated with convection as is saturation fraction). The infrared temperature also shows stark contrast between the convective and non-convective regions. Non-convective regions show a steady high value of infrared temperature characteristic of convection free regions, while the convective regions show a characteristic lag in the infrared temperature minimum compared to the 12 UTC moisture convergence maximum (Bechtold

et al., 2014), associated with stratiform convection which follows deep convection. We notice another maximum in moisture convergence around 19 UTC for convective regions (red line). Flight notes and comparisons of satellite imagery and moisture convergence of the ECMWF operational data show that this secondary maximum is artificial.

Decomposing the convective regions into convective regions in box B2 (blue line), and box B1a (green line), shows that the secondary maximum is a consequence of the model performance in box B2. Convective regions in box B1a agree with our

observations, even showing that the convection in B1a starts earlier in the day (as early as 8 UTC). Box B2, on the other hand, shows an exaggerated maximum at about 19 UTC, not seen in observations, and doesn't show up in the diurnal cycle of infrared temperature as a decrease in infrared temperature as expected from deep convection developing into stratiform convection. We speculate that this secondary maximum might come from overestimating the afternoon saturation fraction and underestimating the instability index by the ECMWF model (blue line, after 15 UTC), in the south part of the B2 box. We hypothesize also that

the satellite wind assimilation issues noted in sections 3.1 and 3.2 might contribute to the model producing spurious convection in box B2, especially in the vicinity of local orographically induced jets which could introduce divergent flow which could negatively influence the satellite wind assimilation algorithms. This is a subject of future research.

In summary, while the saturation fraction, instability index, and DCIN, follow the moisture quasi equilibrium which finds its fingerprints in moisture convergence and infrared temperature, we find an artificial convective maximum in box B2 associated

with potentially overestimated saturation fraction and underestimated instability index supporting this.

## 6 Summary and conclusions

This paper evaluates the impact of dropsondes on the ECMWF model during the OTREC field campaign (Fuchs-Stone et al., 2020; Raymond and Fuchs-Stone, 2021), held during Aug 5 to Oct 3, 2019, in the East Pacific and the Caribbean. The UCAR research aircraft Gulfstream-V performed 22 flights in alternate boxes, dropping a total of 648 successful drops (Vömel et al., 2021). The retrieved fields: zonal and meridional winds, water vapor mixing ratio, and temperature, versus pressure, were assimilated in the operational ECMWF model. To evaluate the model's performance, in addition to using operational data, two experiments are used: with (YDPS) and without (NDPS) dropsondes assimilated. Departures, defined as model values subtracted from observation values, are calculated at the moment of the drop to quantify the deviation of the model from observations for each dropsonde individually.

The vertical departures' mean and standard deviation have small improvements in all fields by inclusion of the dropsondes in the YDPS experiment. The maximum departures of winds in the NDPS experiment occur around 700 hPa, and investigation of the horizontal departure maps shows that the special case of research flight 6 (Aug 18, 2019), a flight into the precursor of tropical storm Ivo, shows the largest departures both horizontally and vertically. Horizontal maps of departures for all variables at 700 hPa (and other levels) benefit from assimilating dropsondes by smaller departures (note the gray areas in the plots, and large departures diminished in the YDPS experiments in figures 2 and 3). The zonal and meridional winds show both vertically and horizontally diminished departures by assimilating dropsondes, while temperature and mixing ratio have small departures in the NDPS experiment. As a consequence, thermodynamic fields used in previous studies, saturation fraction, instability index, and, to a lesser degree, DCIN (Raymond and Sessions, 2007; Gjorgjievska and Raymond, 2014; Raymond et al., 2014; Sessions et al., 2015; Sentić and Sessions, 2017), all show good agreement between the observed and model values. Including dropsondes reduces the spread of the scatter, i.e. increases the correlation coefficient, between the observed and modeled values of these fields. DCIN seems to be overestimated for large absolute values indicating possible boundary layer departures from observations. There is a small difference between the estimated relative vorticity and divergence for the YDPS and the NDPS experiments, with a systematically larger departure of divergence for both experiments. For the special case, box B3, vorticity departures are reduced when dropsondes are assimilated. Departures are small on average as the observing system is well constrained and determined by infrared and microwave satellite observations, and conventional observations, especially over sea (Geer et al., 2017, 2018, 2019). However, the larger departures in the special case (B3 box) when dropsondes were not assimilated, could be caused by weak winds and divergent effects of the local orographic jets from nearby land, which are suspected to introduce errors the satellite wind assimilation algorithms (Bechtold et al., 2012).

Vertically integrated departures for dropsonde denial experiments show smaller departures from observations in cloud free regions, compared to cloudy regions. In cloudy regions, these vertical departures show more scatter, i.e. larger departures from observations, especially in the zonal and meridional winds. The vertical departures of winds show a larger correlation with the infrared brightness temperature, which we use as a measure of cloud cover, compared to the potential temperature and mixing ratio. Water vapor mixing ratio even shows a reduced correlation coefficient with cloudiness when dropsondes are assimilated. The simple definition of cloudiness used in this paper is possibly not sufficient to address the question of assimilation of

moisture (see figure 9c, decrease of correlation coefficient for the YDPS experiment), we leave this question and the question of assimilation of data in the boundary layer to future studies. Temperature and the winds show an improvement, an increase in the correlation coefficient and the slope of the linear fit, especially for large outlier vertical departures, when dropsondes are assimilated (in the YDPS experiment). This suggests that there is room for improvement of satellite derived assimilated winds in cloudy regions, which was also found in previous research (Geer et al., 2019).

To asses the representation of convection and convective parameters in the ECMWF operational analysis with assimilated OTREC dropsondes, we performed an analysis of composites of the diurnal cycle, computed from operational data, for convective and non-convective regions. Instability index and DCIN decrease, and saturation fraction increases, as expected from previous research (e.g. Raymond and Sessions, 2007; Sessions et al., 2015; Sentić and Sessions, 2017), before the onset of convection around 13 UTC. This is consistent with observed decrease of infrared temperature after 13 UTC associated with

stratiform convection of the convective life cycle. A secondary maximum in the ECMWF model moisture convergence, which is associated with deep convection, suggests a second episode of convection around 19 UTC. Satellite observations and our fields notes suggest that this secondary maximum is artificial; further decomposition of the convective diurnal cycle into the East pacific and Colombian box shows that this secondary maximum occurs in the East Pacific box. Large saturation fraction during that period in the south of the B2 box in the ECMWF operational data might contribute to the artificial convection

developed around 19 UTC. The conditions which led to large departures in the special case of box B3 (i.e. low wind speeds and more divergent flows) could play a role in producing this spurious convection in the ECMWF model.

    In conclusion, while the ECMWF model had small departures during the OTREC campaign, for both the dropsondes assimilated and dropsonde not assimilated experiment, there is room for improvement for the assimilated winds, and, to a lesser degree, temperature and moisture fields. Perhaps OTREC data can be used to more specifically address these data assimilation

issues in various satellite assimilation algorithms for the east Pacific region, and extend the findings to other tropical regions. Further ECMWF model study could reveal the sources of anomalous convection in late afternoon, and whether improvements could be made to the assimilated wind in cloudy regions, and regions with low winds or divergent flows. Consequently, further study of the ECMWF model in this region could reveal to what degree we can trust analysis and reanalysis data in the east Pacific region.

*Code and data availability.* Code and model departure data used in this study can be found at doi:10.5281/zenodo.5576132. ECMWF model data in figure 10 is proprietary and the reader is directed to the European Center to obtain the operational data used in this study. The ECMWF IFS operational model is proprietary, so the operational model code is not available to the public. However, An open source version of the code is available at https://www.ecmwf.int/en/research/projects/openifs, where the reader can obtain a licence for the open source code. GOES 16 data was obtained at https://console.cloud.google.com/storage/browser/gcp-public-data-goes-16.

*Author contributions.* Stipo Sentic analysed the data, plotted the figures, and was the main writer of this manuscript. Peter Bechtold performed the ECMWF model denial experiments and participated in the discussion of the results, while Željka Fuchs-Stone participated in data analysis and discussion of the results. Mark Rodwell produced the blacklist files for the NDPS experiment departure calculation, and David J. Raymond participated in the discussion of the results.

*Competing interests.* There are no competing interests present.

*Acknowledgements.* We thank Bruce Ingleby for providing the model data collocated with observational data in the ECMWF assimilation system. We also thank Elias Holm and Mohamed Dahoui for useful discussion and information about the data assimilation algorithms. We would also like thank the two anonymous reviewers for insightful discussion and suggestions. This research was supported by the United States of America National Science Foundation Grant AGS-2034817.

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

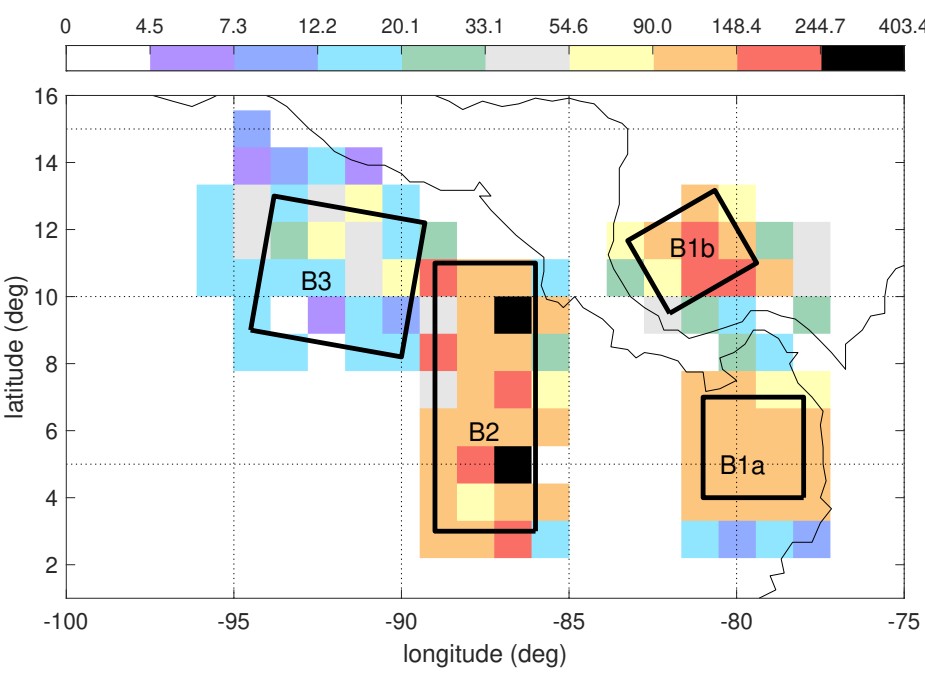

**Figure 1.** Location of the flight boxes in the OTREC field campaign, and density of mandatory pressure level observations used in model assimilation.

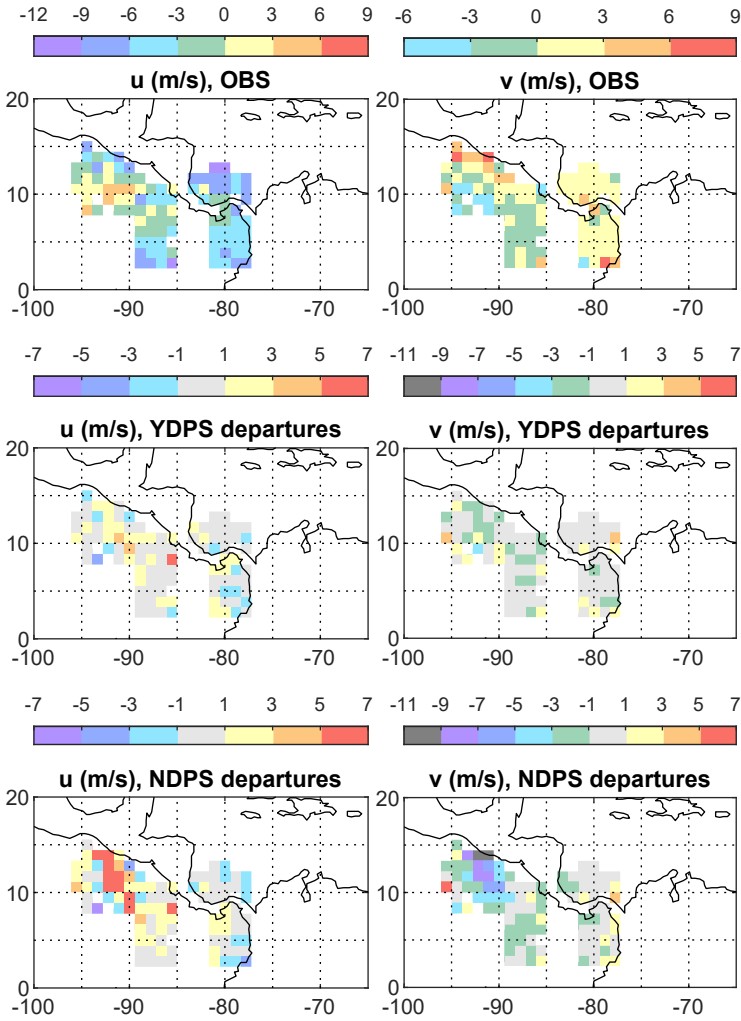

**Figure 2.** The horizontal distribution of mean departures (defined in equation 1) for zonal (left) and meridional wind (right), at 700 hPa. The top row shows mean dropsonde observations, middle row departures of the control (YDPS) experiment, and bottom row departures of the denial (NDPS) experiment.

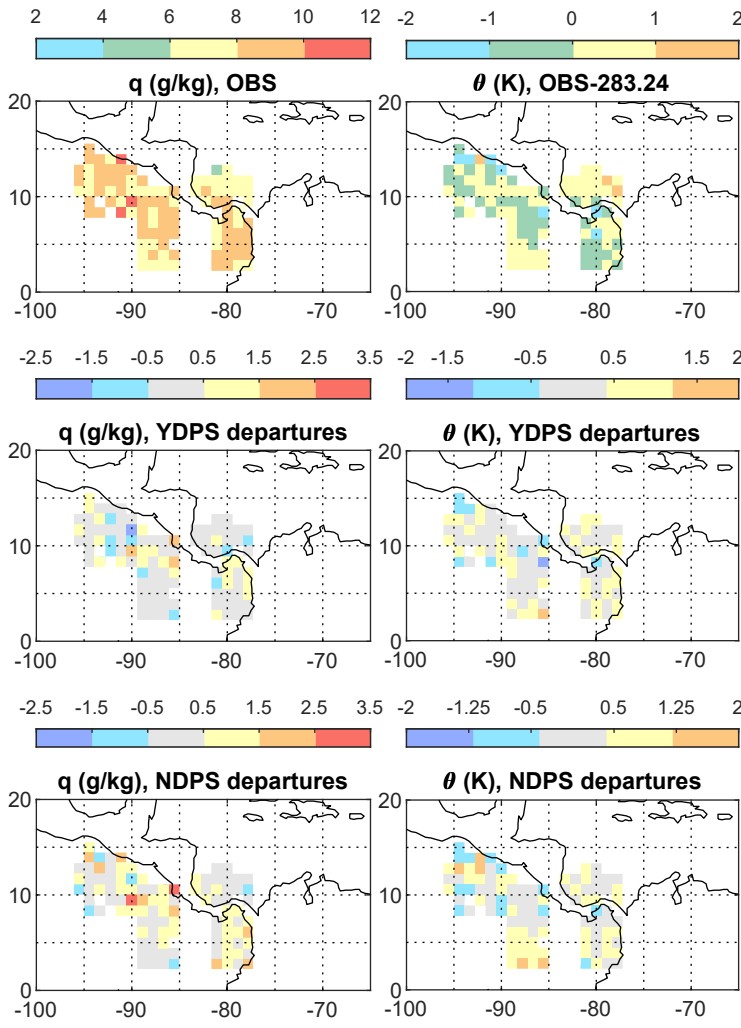

**Figure 3.** The horizontal distribution of mean departures (defined in equation 1) for mixing ratio (left) and potential temperature (right), at 700 hPa. The top row shows mean dropsonde observations, middle row departures of the control (YDPS) experiment, and bottom row departures of the denial (NDPS) experiment.

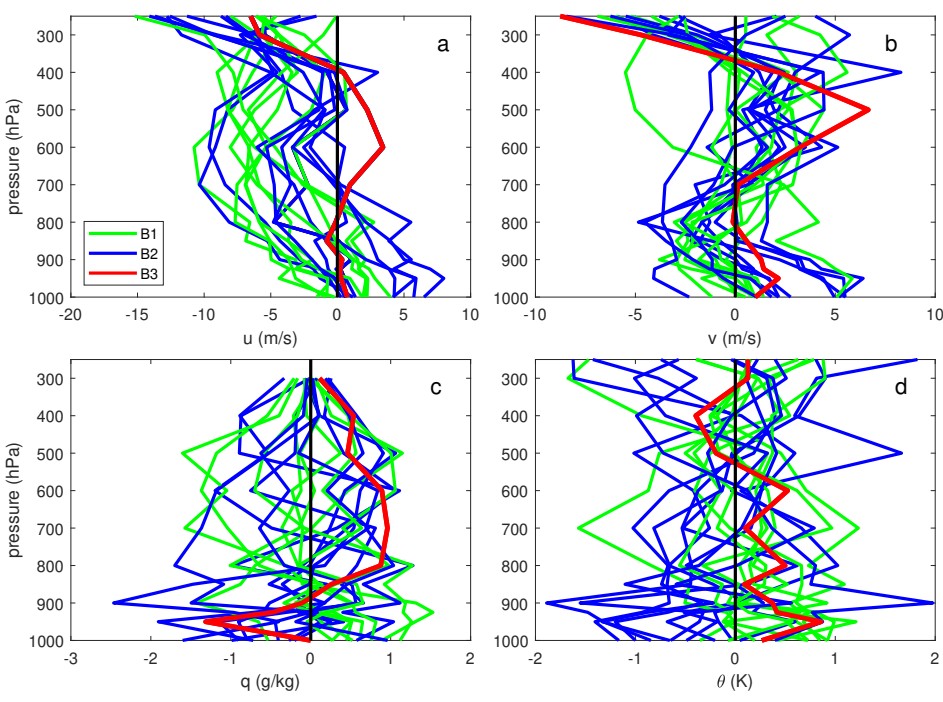

**Figure 4.** The mean dropsonde observations of a) zonal wind, b) meridional wind, c) water vapor mixing ratio, and d) potential temperature, for each individual flight, for the B1 box in green, the B2 box in blue, and red for the special case in box B3. The mean over the whole campaign has been subtracted from mixing ratio and potential temperature.

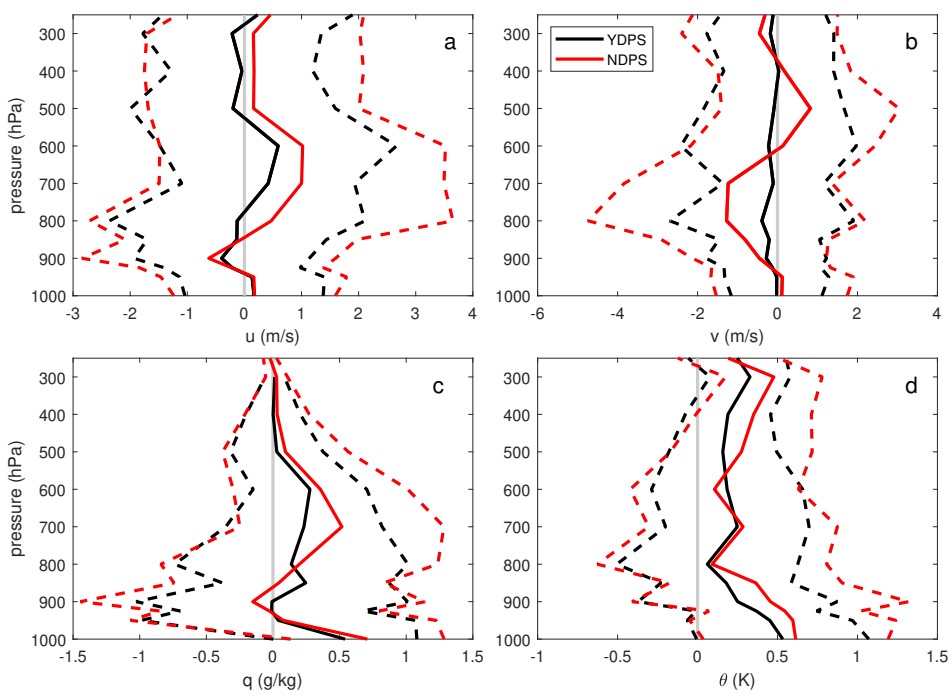

**Figure 5.** The mean (solid line) and one standard deviation (dashed line) of departures from the observations, for a) zonal wind, b) meridional wind, c) water vapor mixing ratio, and d) potential temperature, for the control run in black and the dropsonde denial experiment in red.

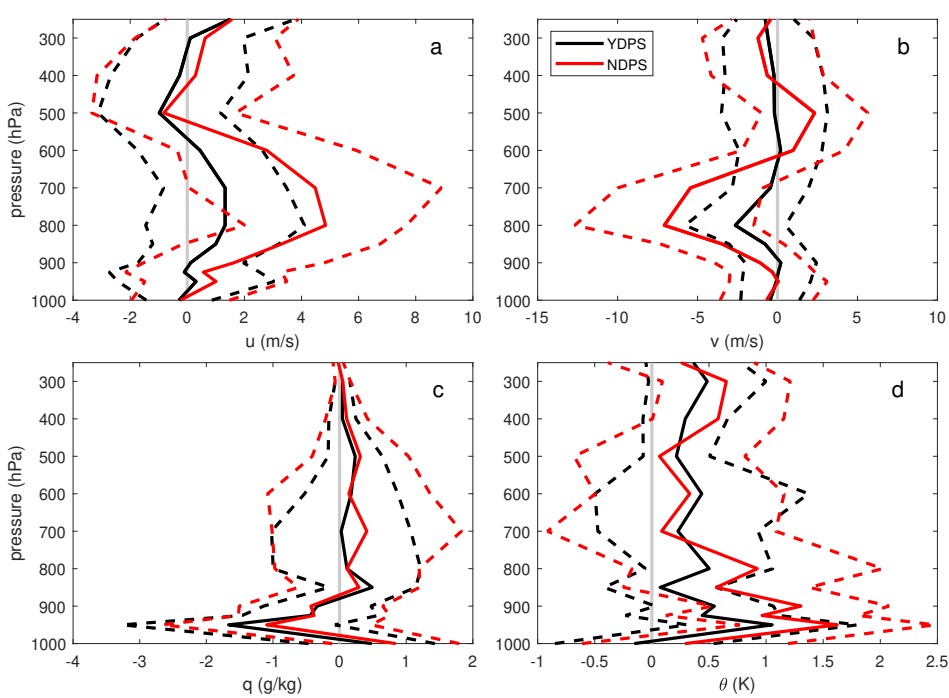

**Figure 6.** Similar to Figure 5, but for the special case in box B3.

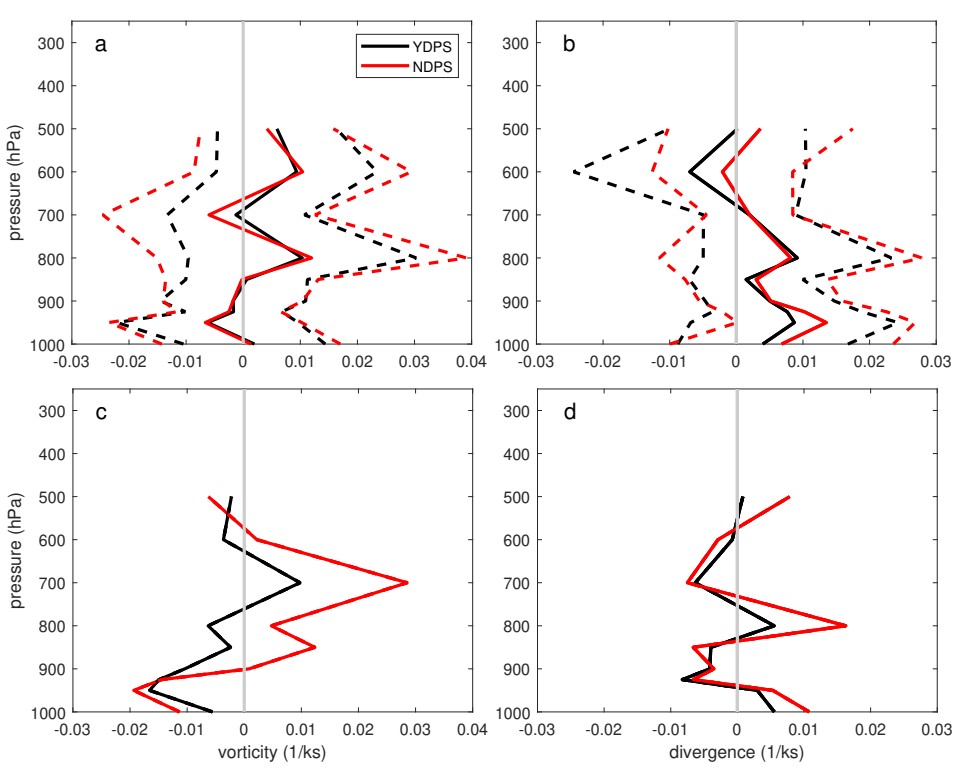

**Figure 7.** Similar to Figure 5, but for a) vorticity departures, all cases, b) divergence departures, all cases, c) vorticity departures for the B3 box case, and d) divergence departures for the B3 box case.

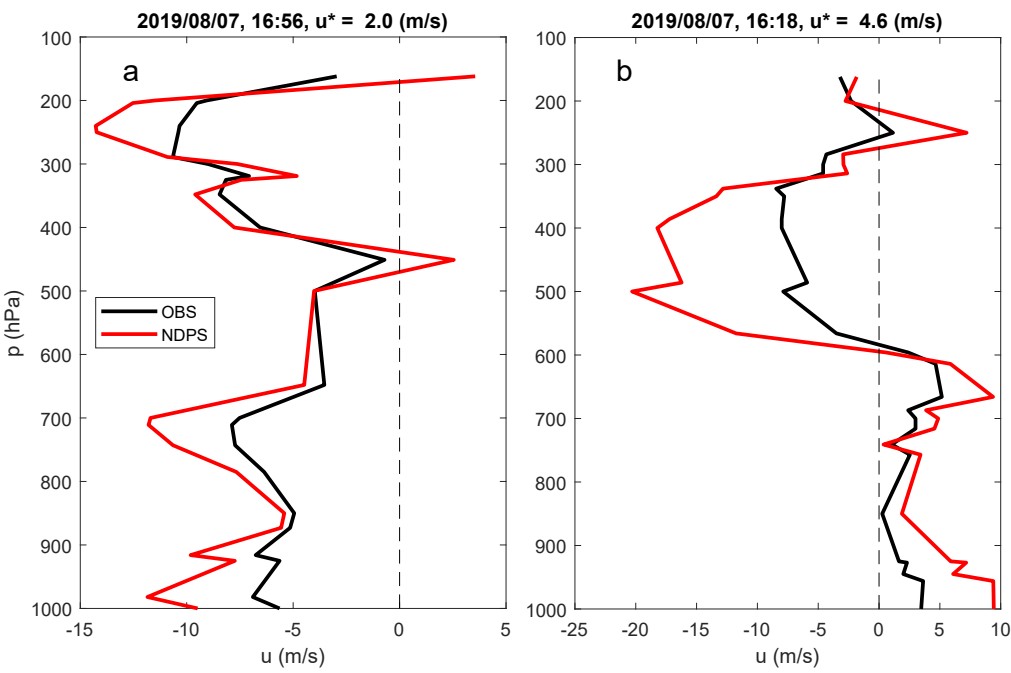

**Figure 8.** Sample zonal wind observed soundings (black) and their NDPS experiment counter part (red). The value of vertically integrated departures $u*$, which quantify how close the model profile is to the observed, are shown above each plot. Panel a shows a zonal wind profile with a smaller vertically integrated departure compared to to the profile shown in panel b.

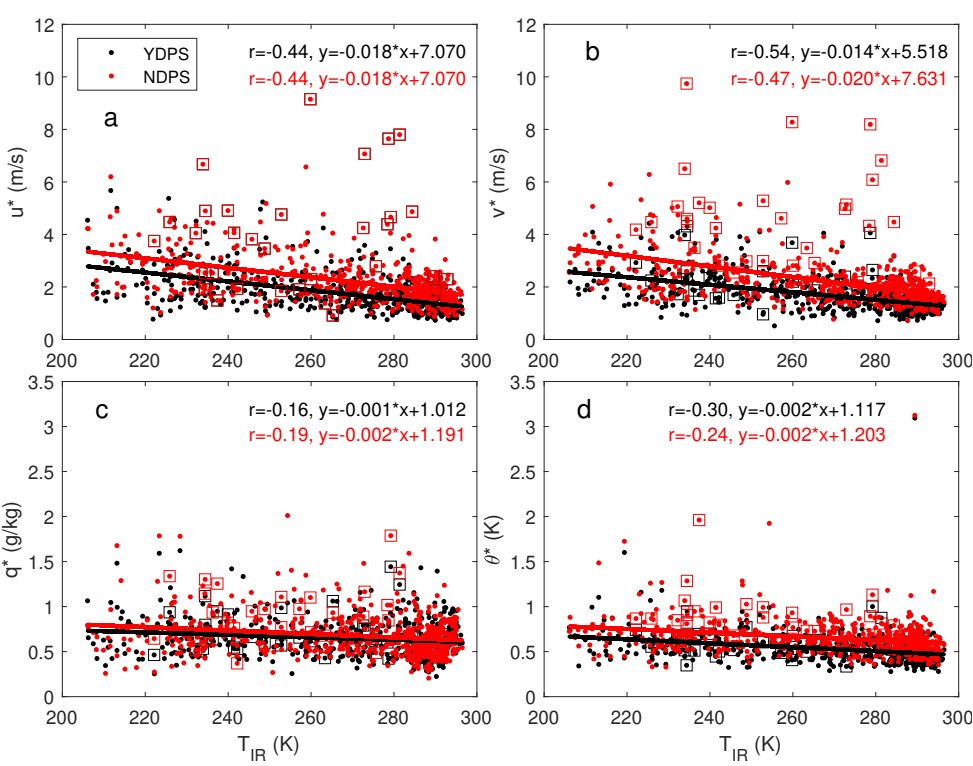

**Figure 9.** Vertically integrated departure of a) zonal wind, b) meridional wind, c) mixing ratio, and d) potential temperature, versus infrared temperature. See text for discussion.

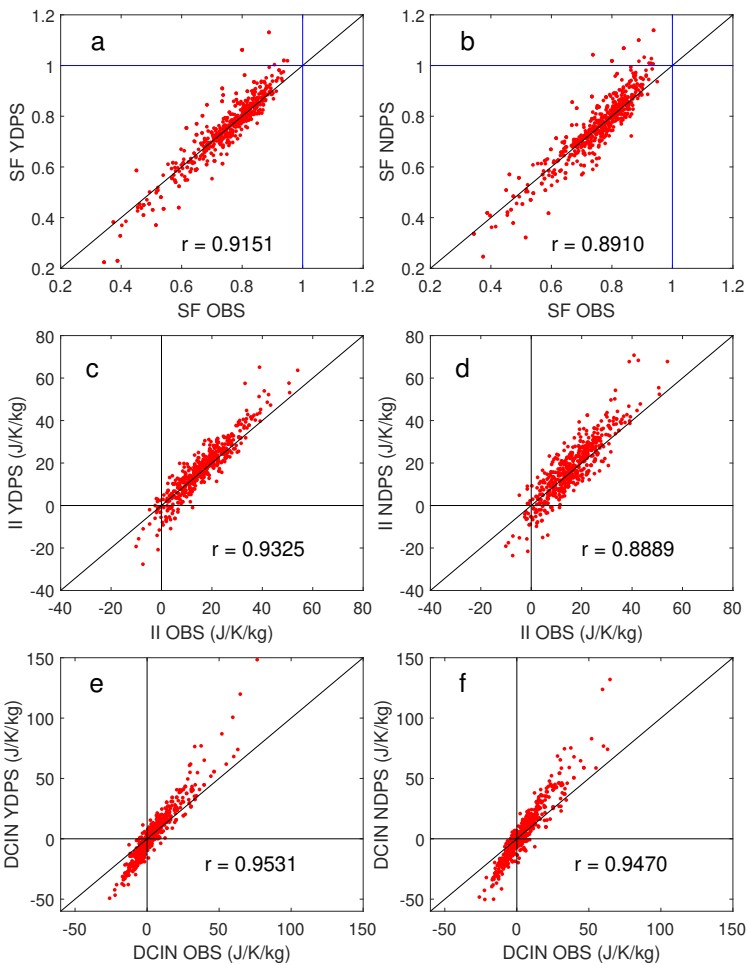

**Figure 10.** a) saturation fraction, observations vs YDPS, and b) saturation fraction, YDPS vs NDPS, c) Instability index, observations vs YDPS, d) instability index, YDPS vs NDPS, e) DCIN, observations vs YDPS, and f) DCIN, YDPS vs NDPS.

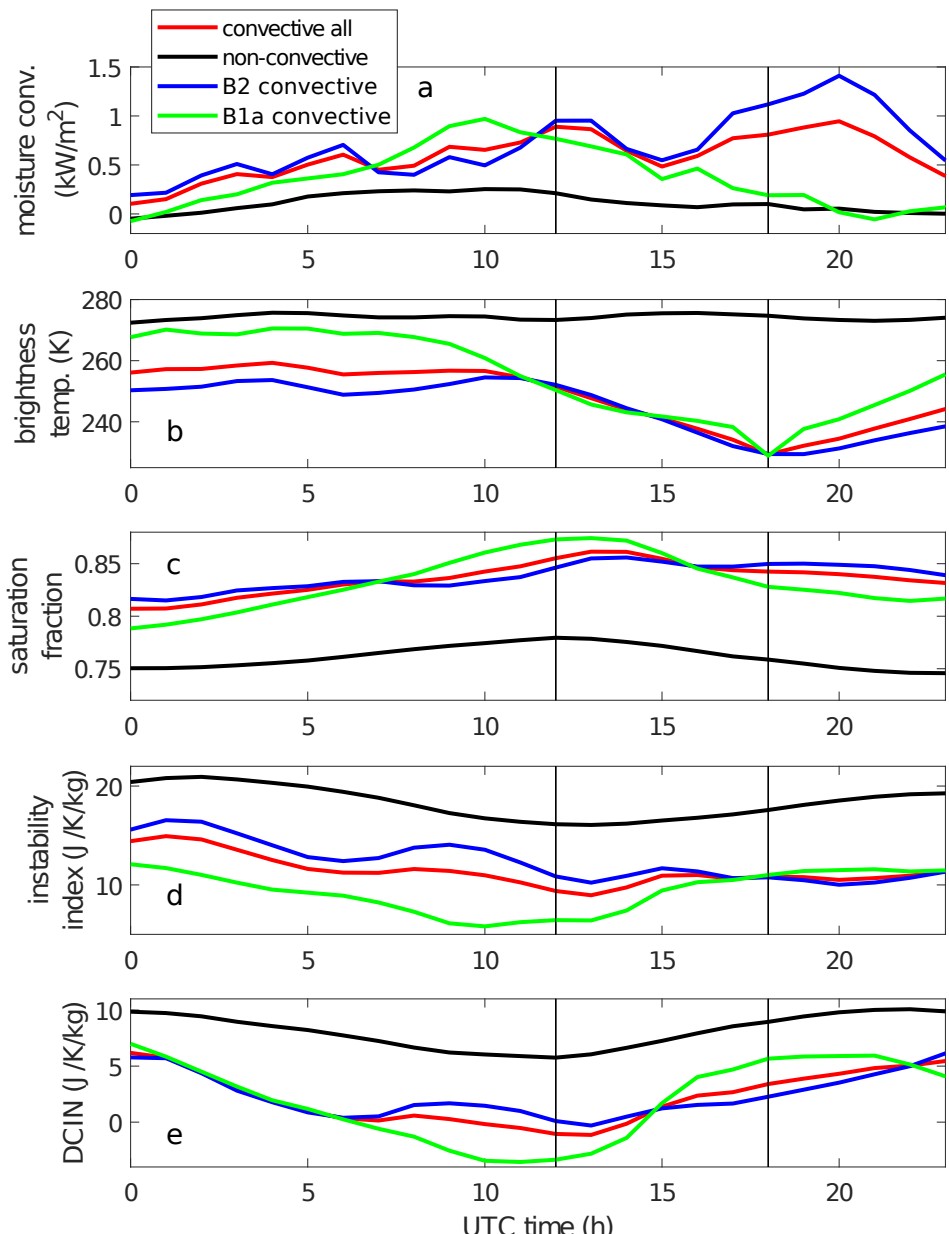

**Figure 11.** ECMWF model time series composites of: a) moisture convergence, b) infrared temperature (from satellite observations), c) saturation fraction, d) instability index, and e) DCIN. Non-convective time series, defined as having moisture convergence lower than 0.5 kW m$^{-2}$, are shown in black, while convective time series, defined as having moisture convergence larger than 2.0 kW m$^{-2}$, are shown in red. Convective time series (red) are also decomposed into the East Pacific box (blue), and the Colombian box (green). Vertical thin black lines indicate the period when the dropsondes were deployed in most research flights.