# Peer review of "On the impact of dropsondes on the ECMWF IFS model (Cy47r1) analysis of convection during OTREC"

_Geoscientific Model Development, 2021_

## Author Comment (AC2)

**Reviewer 1 comments**

Review of the manuscript "On the impact of dropsondes on the EC IFS model (Cy47r1) analysis of convection during OTREC", gmd-2021-354.

General comments:

This manuscript made an overall evaluation of the fitting of the dropsonde observations and the model states with or without these observations assimilated and the influence on convection-related variables. These field campaign data are novel and valuable though the conclusions in this study are intuitive to me. I would recommend a minor revision before it can be published.

 Minor comments:

1. I think the major conclusion from this study is the accuracy of u and v winds are improved by assimilating the dropsonde observations, while the thermodynamic variables has only limited influence, especially the moisture. I think it should be discussed in the conclusion section on the potential approaches to improve the assimilation of moisture variables. Does it mean the critical problem for moisture variables is the deficiency of model physics?
*RESPONSE: Thank you for the suggestion! It could be that the deficiency in model physics gives deficiencies in moisture assimilation. It could also be that the deficiencies in assimilating potential temperature also influences this issue. We discuss this further in the paper.*

2. In Data and methods section, please introduce what the control variables are in the DA system. Are they u, v wind or the stream function and velocity potential? Will the selection of control variables influence the fitting of the u, v wind and the vortex and divergence?
*RESPONSE: Control variables in the DA system are vorticity, divergence, temperature, and relative humidity. We do not think that the selection of control variables will significantly influence the fitting of u or v.*

3. The manuscript has no obvious grammatical problems. However, some descriptions are not clear enough, especially the Data and methods section, for example,

     3. 1 Line56-58, please reword the sentence. Is the resolution of dropsonde observations 13 km? What does it mean "spaced about 1 degree horizontally"? Please

introduce more information about the dropsondes in the data and method section, like the flight altitude, the vertical sampling frequency of the observations.

3.2 Line 79, in Data and methods section, what is the approximate model resolution of the version Cy47r1?

3.3 The model run is from Aug 7 to Sep 30, 2019. How many samples totally are used in this study? What is the output frequency?

3.4 Are all the operational observations assimilated in YDPS and NDPS? Please briefly list their types?

*RESPONSE: Thank you for the suggestions! We expand on all the questions in comment 3, in the manuscript.*

4. Eq. 1, I think it should be the departures of the observations from the model state.
*RESPONSE: We think either way goes. What matters is the correct interpretation of the sign, as noted in the next sentence after the definition.*

5. Lines 88-90, "... this only gives estimates of vorticity departures and not real vorticity departures ...", what does it mean here? Please reword the sentence.
*RESPONSE: Thank you, we clarify this sentence in the manuscript. Since vorticity is not a field for which we can get observations and model values collocated, we estimate the vorticity by gridding the observations and model profiles and then calculating vorticity using finite differences, which of course will not give 100% accurate results.*

6. Lines 136-137, The sentence is not clear, please reword.
*RESPONSE: Thank you for the suggestion! We reword the sentence in the manuscript.*

7. Lines149-150, why do the largest departures for zonal and horizontal wind occur around 700 hPa?
*RESPONSE: That is an excellent question for future work.*

8. The description of Fig. 7 is unclear. It seems that Fig. 7b was not discussed. Why does profile is shown for the experiment NDPS but not YDPS?
*RESPONSE: Figure 7 is an example aimed at showing what a profile of the model and observed value looks like for different x* (where x=u,v,T,q) values. It is not really necessary to show this for YDPS because this figure is not meant as an exhaustive description of different experiments. We do elaborate on this a bit further in the manuscript.*

9. Line 139, which region does Figure 4 show? Averaged over all the regions?
*RESPONSE: Correct, it is averaged over all the data in all the regions.*

10. Please discuss in the conclusion section what potential studies can be done with these valuable dropsonde observations. Can they be used to adjust the model bias though small as suggested in the evaluation?
*RESPONSE: Thank you for the suggestion. We enhance the discussion with your suggestion!*

---

## Author Comment (AC3)

**Reviewer 2 comments**

The authors analyze the impact of a large set of dropsondes over the tropical east Pacific and the Caribbean on ECMWF's integrated forecasting system. Data from two IFS experiments (one making use of the dropsonde data and one not) are compared to study the impact on wind, temperature and humidity distributions. Additionally, the analysis is extended to selected convective parameters which are of relevance for studying developing tropical convection. This impact study is of interest to the community as it makes use of a unique data set and shows a model validation in a region where winds are known to be uncertain. However, it would clearly benefit from additional context and a more thorough discussion of the results. Please see the detailed comments below:

Major comments:

1. The introduction does not provide enough context to the study. It is only hidden between the lines why this work is done and what the questions and hypotheses are. So, I suggest to revise and clarify this section.

*L18: I do not see the distribution of the observations. I recommend that dropsonde locations or frequencies of observations per lat-lon bin (bins shown in Fig. 2) are shown on the map. How many flights were conducted in each box on how many days?
*RESPONSE: Thank you for pointing this out. We have produced the figure 1 map with observation frequency. We also reference an overview paper Fuchs-Stone et al. (2020), with such specific details about flights not directly relevant to this study.*

*L20ff: The reader would profit from information about the general aims of OTREC (instead of detailed results) and how this study is embedded. How were the flights designed? Did they happen regularly or was there a target process, region or time?
*RESPONSE: Thank you for pointing this out. We added a short explanation of the goals of OTREC and the rationale for choosing flights and boxes.*

*L25-32: What is the purpose of the paragraph? Please include a clear statement about the scientific aim of the study and why the IFS data assimilation system and the proposed method is suitable to adress it. Was there a plan for a dedicated validation in the specific regions? Was there a bias expected?
*RESPONSE: Thank you for your suggestion, we have expanded this paragraph with a more accurate statement of scientific aim. Yes, the original OTREC proposal did propose to test model performance in the region, with an expectation of bias.*

*L33ff: I am a bit confused about this paragraph as it covers a broad selection of studies making use of additional sonde data "in other regions of the planet" but it remains unclear what this implies for the presented study. The authors are referred to publications of Majumdar (2016, BAMS) and Parsons et al. (2017, BAMS) that summarize the results of targeted observations during THORPEX and beyond.
*RESPONSE: Thank you for your suggestion. We expand the paragraph to be more clear.*

**2. The data and methods sections would benefit from a more thorough explanation of the model experiments and the used data products. It is confusing to me what type of data is used for what type of analysis:**

*L73ff: This paragraph is a surprising start as this 3D-var approach was never mentioned before and it is not stated why it is needed.
*RESPONSE: We added the rationale for using the 3D-var in this study. We use it for calculating mass balanced fields derived from the basic observed fields (zonal wind, meridional wind, temperature, and mixing ratio). The derived field which we use to identify convective and non-convective regions, in section 5, is moisture convergence.*

*L79-84 (and elsewhere): Please specify "model data" and what is used in the departure calculation. Is it an analysis departure or a background (first guess) departure? Please explain how this can be interpreted? Why do only positive values denote a deficit? Please better explain the two experiments (YDPS and NDPS) and what one expects to learn from comparing them? Were these experiments cycled over the entire month or several single cycle experiments for each mission? Please give details about the blacklisted profiles mentioned in L166. I guess the analysis only considers data points that are used in both experiments, correct?
*RESPONSE: I am a bit confused with the reviewer's comment; the variables used for calculating departures are listed in the same paragraph, a few lines further.*
*We use analysis departures because those are what end users deal with when they receive the data from ECMWF, that is why we do not look at first guess departures.*
*The way we defined departures: observation minus model value, means positive values denote model deficit relative to observations. It was not a judgement on the results, it is just so by definition. We made that clear in the revised manuscript.*
*We better explain the experiments at the expense of being repetitive in the paper.*
*We also add details you requested; that the model was cycled over two months, and yes, the analysis only considered data points used in both experiments.*

*L83: I guess there is no collocation done in the sense that data is interpolated in space and time, but observation space data is used from the data assimilation system? Is the restriction to significant level data a result of the transmission of thinned data to the GTS? Please explain what data was used and clearly separate observation space data

from gridded model data.

*RESPONSE: Significant levels: The OTREC radiosonde data are high resolution data, but due to some unfortunate technical issues, the data assimilation system could at the time not assimilate them as high resolution BUFR data, but only on significant levels, resulting in some loss of information.*

*L102: Here, I guess that "operational data" means gridded model space data? Do the 1x1° cells mean that data is retrieved at this grid resolution? Is the operational analysis data used here? What is the vertical resolution? L108 says that also ERA5 data is used but it is unclear why? Please be more precise about the data.

*RESPONSE: Thank you for pointing this out. We corrected this paragraph with correct data usage. We use the 3d var analysis data to get the locations of convective 1 by 1 degree boxes from observations. The location of these convective events is then used to get diurnal cycle data from operational gridded data from which the diurnal cycle mean is calculated. The 3d var data is calculated on a 0.25 by 0.25 degree grid and then averaged into 1 by 1 degree data.*
*Thank you for pointing out the ERA5 discrepancy. In an earlier iteration of the paper we used both operational and ERA5 model data, but since the difference was not big we decided to use the operational data and left the ERA5 data in by mistake. We have corrected this error now.*

3. The section "model analysis" that contains the main results to assess the impact of the dropsondes is well-written and good to follow. The authors identify one particular flight on Aug 18, 2019 providing increased departures which is very interesting. However, it remains unclear whether this is a special case in terms of the observed profiles (stronger wind, more humidity or warmer?) compared to the other flights and also whether the observed pre-tropical storm environment differs from other flights.
*RESPONSE: Thank you for the suggestion. We expand this section with more information about how this special case differs from other cases. Please see below (*L125 comment) on how we chose the special case.*

I would like to suggest that the authors enhance this discussion. Difference maps between the analyses of both experiments (or analysis and first guess) could help to understand how the information contained in the dropsondes was spread in the domain. Additionally, the largest departures could be illustrated with respect to the synoptic situation. Did the assimilation also improve the later development of the tropical storm?
*RESPONSE: Looking at the influence of the data assimilation on forecast of the tropical storm is beyond the scope of this paper, but excellent suggestion for future work!*

*L118ff: This discussion would profit from showing how many observations were made in each box. How many flights contributed to the result in Fig. 2? See also comment on L18.

*RESPONSE: Thank you for the suggestion. We enhance figure 2 with frequency of observations. We also list in the introduction how many flights covered each box.*

*L125: I guess "statistically significant" means that in the other boxes, departures from more than one flight are averaged which might have led to reduced departures?

*RESPONSE: At that location in the paper we refer to the following figure.*

[Figure]

*This is a figure 4 equivalent, averages of departures for each individual flight, black YDPS and red for NDPS. The outlier red line obvious in 'u' and 'v' is the NDPS experiment for the research flight in the B3 box. This shows that the B3 box, the special case in the paper, is statistically significantly different from the other flights in the wind field.*

*L128f: "Lack of... " I do not understand this sentence. What does this study tell about a negative impact on the prediction? Does that mean that other obs types are not able to constrain the analysis correctly?

*RESPONSE: Thank you for pointing this out, we endeavor to make the explanation more clear. We infer that the prediction might suffer in such a case as this because, in absence of a OTREC field campaign, the initial condition would predict a storm of different characteristics. We can see how the sentence is confusing and reworded it. Also, yes, the NDPS experiment, which uses other observations to constrain the initial*

*conditions, does imply that those other types are not able to constrain the analysis correctly when compared to dropsondes.*

*L131: "This suggests…" If so, are there other examples or might that be just very case sensitive? This also affects the next argument on the sensitivity to the boxes. Was this flight designed differently or focusing at a different met. situation? Was it the only pre-storm flight? Was this case unique, e.g., in terms of wind speeds? See also comment on L20ff.
*RESPONSE: You could be right. The special case might reflect different conditions as the mission there was targeting a pre-tropical storm disturbance environment. Here we plot the average values of the basic variables for comparison. As we explain above, this case was statistically significantly different from other departure cases. We do discuss this in more detail as you suggest above.*

*L143: I do not understand this sentence. What is the relative error?
*RESPONSE: Relative error is defined as |(X-Y)/X|*100%, where X is the observed value, and Y is the model value.*

*L156: What does "issues" mean? Were large FG departures leading to a withdrawal of the data? Is the used data rejected during the data assimilation process?
*RESPONSE: We rewrote the sentence to be more clear. Actually no, the YDPS experiment did assimilate data so that is proof that the data was not withdrawn or rejected. The small departure in the YDPS experiment for the special case shows that the data had enhanced weight during the assimilation.*

*L169: Please explain more carefully how this is done as it remained unclear what kind of data is used. Please specify "point data to a regular grid". Is this observation space or model space data? Is this related to the methods briefly introduced in the first paragraph of section 2? I guess this method uses observation data and the two analyses from the experiments? A bit more information is needed for readers that are not aware of results shown in Fuchs-Stone et al. (2021). What if the results were compared with gridded model space data? Is it possible to derive reliable vorticity data from the relatively coarse observation grid?
*RESPONSE: Thank you your suggestions. We rewrite "point data to a regular grid" to "linearly interpolating the point data to a regular grid". We do this both on the dropsonde data and on the model blacklisted data, i.e. we treat them as model "dropsondes". This is not related to the 3d var analysis described in section 2. Since the ECMWF model has parameterizations that influence the divergence and therefore mass flux profile calculations we found it was not possible to compare the 3dvar divergence and vorticity directly. Therefore we resort to estimating the divergence and vorticity that would be ostensibly assimilated in the model.*

*Since the paper Fuchs-Stone et al. (2021) is open source we do not see a problem in leaving the reader to check Table 1 of that manuscript.*

4. The section on cloudiness effects is entirely convincing to me, primarily as I missed a hypothesis about what is expected and why this is done. I do not get why x* should be higher in clouds especially for winds. The presented analysis doesn't tell anything about whether a cloud also exists in the model and does not consider the vertical extent of the clouds and cloud layering. Why did the authors not consider a point by point analysis and use something like a dew point depression or RH as an indicator for clouds? Could it be that highest departures occur where no cloud was simulated but observed or vice versa?

*RESPONSE: It is not that x* should be higher in clouds, it IS higher in clouds. That is, that correlation is observed, not inferred. It is not the point of this paper to check if there is a could in the model, and also the goal is not to check for the vertical extent of the cloud. We elaborate in the revised manuscript what the goal was; since the ECMWF data assimilation system uses global irradiance values, i.e. satellite data, for assimilation, it seems natural to use satellite data as a measure of presence or absence of clouds. Presence of clouds will clearly affect the assimilation of irradiance data. We have failed to make this clear in the manuscript so we enhance the discussion in this section of the paper to justify our approach.*

*L223: I do not understand this sentence as I thought the difference between the experiments is only the information contained in the dropsondes (plus a potential cycling effect that should also be discussed in the final section)? Are there other obs types that do not perform well? What types of wind observations do exist in clouds?

*RESPONSE: Thank you for pointing out that this sentence is unclear. We expand the manuscript to be more clear. Indeed, the assimilation system does not only use dropsondes, but in both cases there is other data assimilated. It is beyond the scope of this paper to study details of which other types of data affect the assimilation.*

*L232: The Instability Index seems also to show some overestimation in both experiments. Although this is also the case for DCIN it is stated that the convection parameters are well-represented. Later departures are considered "small" and an "excellent agreement" is seen. What is the underlying accuracy requirement for these statements? I guess a 1 g kg$^{-1}$ or a 1 K might strongly impact a process like convection.

*RESPONSE: Thank you for pointing this out. High values if Instability Index, and negative values of Instability index are not as important as the middle range; the most important physics of tropical convection happens in that range. We expand our discussion of the impact of the spread of the scatter between observed and modeled values. That is correct, 1 g/kg and 1 K can impact convection significantly as shown in Raymond and Sessions (2007).*

5. I asked myself what section 5 has to do with the topic of this paper? What is the reason for adding an analysis of the diurnal variability of these parameters in ERA5. It would be more interesting to know to what extent the assimilation of dropsondes contributed to the diurnal cycle representation in the forecasts, e.g., when using the 12 UTC FCs over 24 h and compare it to the analyses.

*RESPONSE: As noted above, I erroneously left "ERA5" where "operational analysis" is supposed to go. We correct that in the manuscript. We do not go into the proposed analysis because of the nature of our experiments. Since we ran a two month cycle instead of running the model 22 times at the moment of each research flight, there is stochastic differences occurring which we would not be able to distinguish from dropsonde influences. Therefore we look at average properties by constructing the diurnal cycle. There is sufficient discrepancy between the observations and model performance in the diurnal cycle to suggest further work necessary to address such questions.*

*L236: Why is ERA5 data used here instead of the experiments? Please clarify how observations and simulations are combined and whether this 3D var method is applied to the ERA5 gridded data to calculate a time series of convective parameters.

*RESPONSE: Thank you for pointing this out. We clarify this in the manuscript. We do use the operational analysis and not ERA5, as noted in a previous response.*

*L237: The separation of non-convective vs. convective regions in the domain is done only for the dropsonde releases, right? It would be good to give an overview about when the flights are conducted and dropsondes are released in the overview section. Does the difference in release time lead to an average moisture convergence below 2 kW m$^{-2}$ as shown in the time series? How many flights or individual days did contribute to this analysis?

*RESPONSE: The 3d var analysis was done on the days of dropsonde releases, correct. We assume that the differing times of releasing dropsondes minimally influences/distorts the analyzed fields because lawnmower patterns done during the flights were designed to minimize those effects. The moisture convergence should not be affected drastically by the different dropsonde release times. There were 22 individual flights on 22 separate days.*

*L260: The decomposition is interesting but to understand the inter-box differences it would be good to know how many days are investigated here.

*RESPONSE: In line 244 we list how many cases are used for calculating each average. We do not find it necessary to distinguish the number of days over which the average was taken because the flights were performed at the same time, 12-18UTC each day.*

**6.** The **summary and conclusion** section misses a discussion of the results in the context of previous literature and the aims of this study. Although the study addresses the IFS model performance by comparison of model results with observational data, I wonder why the authors submitted the paper to GMD. Therefore, I suggest that the authors comment on how an NWP center can profit from these results. I would also encourage the authors to discuss their results in the context of the quality of tropical winds.

*RESPONSE: Since we discussed the results in the context of each section of the paper, we summarized the discussions in this section. We expand the summary though, to include the main discussion points from each section.*

*We chose GMD for the manuscript because the paper highlights performance issues of the ECMWF model which the modelers can further explore in development of the model in the East Pacific region. Since there has not been an observational study in the East Pacific region since the field campaign EPIC in 2001. This study offers unprecedented amount of data in the region used in the study.*

**Minor:**

*Title: I suggest using "ECMWF" instead of "EC"
*RESPONSE: Thank you for the suggestion. Suggestion was adopted in title.*

*L5 and L59: Please check the abbreviation "EC" as typically ECMWF is used.
*RESPONSE: Thank you for the correction.*

*L9: Please specify "precursor"
*RESPONSE: Thank you for the correction.*

*L16-17 and elsewhere: correct citation brackets
*RESPONSE: Thank you for the correction. We corrected the brackets throughout the paper!*

*L29: reanalyses?
*RESPONSE: Thank you for the correction.*

*L37: Please specify "affected"
*RESPONSE: Thank you for the correction.*

*L105: What is the top pressure? The uppermost model level?
*RESPONSE: The top pressure we used here is 200 hPa.*

*L109: Please find a different and more specific title.
*RESPONSE: Thank you for the suggestion.*

*L113 i.e. 700 hPa?
*RESPONSE: Thank you for the correction.*

*L121: "all" seems not to be correct.
*RESPONSE: Thank you for the correction.*

*L123: Box 3?
*RESPONSE: Correct, as noted in the next sentence.*

*Fig.4: At what vertical resolution is this analysis performed and what is the data availability? Use Theta symbol for potential temperature instead of t.
*RESPONSE: We perform it on standard levels even though data is available on more pressure levels. Since the pressure levels do not coincide between different locations we average data on standard levels and perform the analysis there. We changed t to theta in the figures, thank you.*

*L230: Please specify "considering the previous section"
*RESPONSE: Thank you. Expanded.*

*Fig.8 and Fig.9 would profit from a fit to the data (Fig.9) and some statistical numbers
*RESPONSE: Thank you for the suggestion. We added correlation coefficients and other measures of fit.*

*L272: "evaluates the performance" isn't it rather an evaluation of the impact of sondes as stated in the title.
*RESPONSE: Thank you for the correction.*

*L282: "variable departures" is ambiguous
*RESPONSE: Thank you for the correction.*

*L284: Please rephrase "benefit by … smaller departures"
*RESPONSE: Thank you for the correction.*

*L293: Change to "more scatter, i.e. more variable departures"
*RESPONSE: Thank you for the correction.*

*L297: please mention that this was done using ERA5 data
*RESPONSE: Thank you for the correction. We corrected that operational analysis was used.*

*L307: "whether improvements could be made"
*RESPONSE: Thank you for the correction.*

---

## Author Response (AR2)

**Response to reviewer #2, second review**

This is my second review of "On the impact of dropsondes on the ECMWF IFS model (Cy47r1) analysis of convection during OTREC". The authors followed several of my concerns and I would like to thank them for the additional explanations. The revisions and revised and new Figures certainly led to an improvement of the presentation of scientific aims and data and of the discussion of the results.

The special case is much better integrated now. The paper would befit from a connection of the special case to the findings about the cloudiness effects which would be easy to realize. The authors could show whether the profiles provide high or low brightness temperatures and how this relates to a good use of the data and a particularly strong improvement.
*Response: Thank you for the suggestion. I have updated the figure 9 (brightness-vertical departure plot) to mark the 32 points from the special case. Indeed, they shop up as the majority of the largest departures in the NDPS experiment. We updated the section 4 with the discussion of these highlighted points.*

I still feel that section 5 is detached from the rest of the paper and I would have preferred more details about the special case, but these results seem to be connected with earlier work by the authors, so I understand why the section was kept.

The authors remain very vague about how the results help to improve algorithms. How can the revealed indications for an insufficient use of winds in cloudy situations be used? I guess this should be universal not only to the eastern Pacific that the highest impact is expected in low wind and divergence situations and affected by clouds?
*Response: Thank you for pointing this out. We find it is beyond the scope of this paper to address these questions in detail, and leave such work to future studies.*

When reading the paper again I found several mostly minor issues that the authors should consider:

L30f: This sentence could be deleted or write "analyses".
*Response: Thank you for the suggestion. We summarize the "operational analysis and reanalyses" with "analyses".*

L35f: Should be Agusti-Panareda et al. (2020)
*Response: Thank you for the suggestion, but we do not find that to be correct. The reference is from 2010.*

L45f: This sentence contains no information, neither about the models nor about the content of the reference.
*Response: Thank you for the suggestion. We specify that the authors used dropsonde assimilation (denial) experiments.*

L47: What is "accuracy of the outflow (analysis?)"?
*Response: The authors in that reference analyze outflow related parameters. We specify this in the manuscript.*

L61: "on the ECMWF analysis and tropical convection" sounds odd.
*Response: Thank you for the suggestion. We reword this to "on the ECMWF analysis and modeling of tropical convection".*

L66ff: Please rephrase this sentence, e.g. delete "those parameters". Rethink the position of the references – wouldn't they fit better after "characterizing convection"?
*Response: Thank you for the correction. Implemented!*

L99: I could not find the word "suficit" in common dictionaries. I guess you mean an overestimation? Here, I also suggest to mention that a reduction of the departures after assimilating the dropsonde data means that the model is able to make use of the data and that errors are reduced.
*Response: Thank you for the suggestion. We reworded the clumsy wording to "overestimated and underestimated" instead of "deficit and suficit", and added a sentence to the suggested effect.*

L100: "Model levels are interpolated to the location": As I already suggested during the first review, please specify whether this is done within the 4D-VAR or by interpolating the gridded-fields. If the latter is true: 1) How is this done? (linearly in space, in time) 2) What errors do you expected?
*Response: Thank you for reiterating this point. It was done with the 4D-VAR and we indicate this in this sentence.*

L104ff: Not sure why this paragraph was added and is needed.
*Response: Ah, yes. You suggested in your previous review to put in which other fields (other than dropsondes) are used for data assimilation. We still find this useful for the reader to be aware of all the other fields assimilated in the model and leave the paragraph as is.*

L135f (and 309ff): What data is exactly used here? Operational analysis data is available only every 6 hours. Did you include FCs and when were they initialized?
*Response: Thank you for pointing this out. We use the forecast initialized at 12 UTC.*

L151: The reference Ivo (2019) cannot be correct.
*Response: It is a reference to the year of the storm Ivo; 2019 hurricane season. We emphasize this.*

L149: Please delete "in the placement of convective systems" as you never get to this point
*Response: We replace it with "modeling of convective systems".*

L163f: This is very vague, it might also be the case that the available data wasn't used properly and it is not clear if this holds for more than one case.
*Response: We add "for all other individual flights". We shared the figure with the reviewer in the response document, but find it would clutter the manuscript so we omitted it. If the reviewer refers back to our response they will find this figure:*

[Figure]

*which clearly shows our point (note the statistically significant outlier in panels and b).*

L179: "which certainly seems to happen here" This may be correct but it is not investigated so that I recommend to weaken this conclusion.
**Response:** *We deleted "certainly" to weaken the conslution.*

L189: I guess you mean "As absolute differences these departures are …"
**Response:** *Correct. Corrected. Thank you for the suggestion.*

L194: Why "official"? I guess you mean wind errors? Be more specific.
**Response:** *Thank you for pointing this out. We put "official" because the reference is not peer reviewed and we wanted to give it more weight. We specify wind errors at the end of the sentence.*

L205: I guess "divergent conditions" does not mean increased divergence but rather the differences between the departures? Please be specific about what you mean here. I cannot see a local jet in Fig.4. Please add a reference to the phenomenon.
**Response:** *We do mean increased divergence. We make that clear in the manuscript. Indeed, we do not show the jet in this plot but because of the climatology of the region we assume it might influence the OTREC flight regions. We add references for the jet climatology.*

L225f: What was used in case of the model data? Was the data taken from one particular time step or temporally interpolated?
**Response:** *We used fields necessary to compute divergence and vorticity; winds and positions of dropsonde drops. As we note in the manuscript, we assume the drops were made at the same time for*

*the purposes of the calculation.*

L263ff: What is the TIR of both soundings and where are they located in Fig. 9. Does the special case also provide a special behavior in this analysis (see remarks at the beginning)? This could be a nice way to connect these separated topics.
*Response: The reason why we use these two examples is to show the definition of the vertically integrated departure. At this point in the section we do not correlate it with TIR. But you are right, we do connect this section with the special case (see comment at the beginning of the review); we draw squares around the special case points in figure 9, and discuss this in the section. Indeed they show that the special case had larger departures in wind, and were corrected with inclusion of dropsondes.*

L277f: "q* and T* seem to perform well" makes no sense to me. Do you want to say that errors are the same inside and outside clouds while for winds the scatter increases and errors are higher?
*Response: Thank you for the suggestion, we have corrected this in the manuscript.*

L280f: I thought it is one experiment.
*Response: As figure 9 shows we have two experiments, the NDPS and the YDPS experiment (see description in the methods section).*

L281: Generally, the correlation is low which should be discussed.
Regression lines in Fig. 9 are hard to see.
*Response: Thank you for noticing this. We discuss the small values of the correlation coefficients in this paragraph.*

L289: Take care that units are correctly typeset when using the format "J/K/kg"
*Response: Thank you, we took care of the latex typesetting but await further corrections from the technical editorial staff from the journal.*

L356: "maximum departures of winds"?
*Response: Thank you for pointing this out. Corrected.*

L358: Horizontal maps don't benefit
*Response: We disagree. Although the departures don't seem to benefit due to the plotting boundaries for each color, the vertical departure improvements do reflect on the horizontal map improvements. Its subtle but still present in the horizontal maps.*

L359: Please be more specific why you consider these departures to be small? Shouldn't it be mentioned that these departures aren't corrected through dropsonde assimilation.
*Response: As we can see in the figures 2 and 3, the fractional area of "gray areas" (smaller departure) is larger. Also, large departures (red and yellow) become smaller (gray areas). So, as mentioned in the previous line response (above), While it is subtle, we do think there is improvements by assimilating dropsondes.*

L365: Rephrase "spread of the scatter"
*Response: Thank you for pointing this out. Corrected.*

L373: More scatter is not necessarily meaning larger departures
*Response: Actually, in this case it does, as it is shown with the slope of the regression line in figure 9, for cloudy regions.*

L376: please rephrase "quantifies" as this rather an indication for clouds and also the representation of clouds in the model is not treated.
*Response: Thank you for pointing this out, we rephrased it as "which we use as a measure of cloud cover".*

L379: What "question of assimilation of moisture"?
*Response: We refer to figure 9c. There is decrease of correlation for the YDPS experiment. We clarify this in the manuscript.*

L380: Please clarify this sentence: The improvement is not only seen in the correlation.
*Response: Thank you for pointing this out. We clarified this in the manuscript.*